# A remodeled RNA polymerase II complex catalyzing viroid RNA-templated transcription

**Shachinthaka D. Dissanayaka Mudiyanselage[1], Junfei Ma[1], Tibor Pechan[2], Olga Pechanova[2], Bin Liu[1], Ying Wang[1] ***

**1** Department of Biological Sciences, Mississippi State University, Mississippi State, Mississippi, United States of America, **2** Institute for Genomics, Biocomputing and Biotechnology, Mississippi State University, Mississippi State, Mississippi, United States of America

* wang@biology.msstate.edu

## Abstract

Viroids, a fascinating group of plant pathogens, are subviral agents composed of single-stranded circular noncoding RNAs. It is well-known that nuclear-replicating viroids exploit host DNA-dependent RNA polymerase II (Pol II) activity for transcription from circular RNA genome to minus-strand intermediates, a classic example illustrating the intrinsic RNA-dependent RNA polymerase activity of Pol II. The mechanism for Pol II to accept single-stranded RNAs as templates remains poorly understood. Here, we reconstituted a robust in vitro transcription system and demonstrated that Pol II also accepts minus-strand viroid RNA template to generate plus-strand RNAs. Further, we purified the Pol II complex on RNA templates for nano-liquid chromatography-tandem mass spectrometry analysis and identified a remodeled Pol II missing Rpb4, Rpb5, Rpb6, Rpb7, and Rpb9, contrasting to the canonical 12-subunit Pol II or the 10-subunit Pol II core on DNA templates. Interestingly, the absence of Rpb9, which is responsible for Pol II fidelity, explains the higher mutation rate of viroids in comparison to cellular transcripts. This remodeled Pol II is active for transcription with the aid of TFIIIA-7ZF and appears not to require other canonical general transcription factors (such as TFIIA, TFIIB, TFIID, TFIIE, TFIIF, TFIIH, and TFIIS), suggesting a distinct mechanism/machinery for viroid RNA-templated transcription. Transcription elongation factors, such as FACT complex, PAF1 complex, and SPT6, were also absent in the reconstituted transcription complex. Further analyses of the critical zinc finger domains in TFIIIA-7ZF revealed the first three zinc finger domains pivotal for RNA template binding. Collectively, our data illustrated a distinct organization of Pol II complex on viroid RNA templates, providing new insights into viroid replication, the evolution of transcription machinery, as well as the mechanism of RNA-templated transcription.

## Author summary

Viroids often cause disease in crop plants and represent the simplest form of pathogens that contain genetic materials. Viroid genetic materials are made up of small, circular-form, single-stranded RNAs that do not contain any protein-coding genes. To make

**Data Availability Statement:** The raw mass spectrometry proteomics data have been deposited

to the ProteomeXchange Consortium via the PRIDE partner repository with the dataset identifier PXD033736. The organized mass spectrometry proteomics data of each replicate are summarized in S1–S6 Datasets.

**Funding:** This work was supported by US National Science Foundation (MCB-1906060 and MCB-2145967 to YW), US National Institute of General Medical Sciences (1R15GM135893 to YW), and NIH MS-IDeA Network of Biomedical Research Excellence award 5P20GM103476-19. The mass spectrometry proteomics analysis was performed at the Institute for Genomics, Biocomputing and Biotechnology, Mississippi State University, with partial support from Mississippi Agricultural and Forestry Experiment Station. The funders had no role in study design, data collection and analysis, decision to publish, or preparation of the manuscript.

**Competing interests:** The authors have declared that no competing interests exist.

progeny copies, viroids that replicate in the nucleus redirect host DNA-dependent RNA polymerase II (Pol II) to work on their RNA genomes. This process also occurs with the human hepatitis delta virus when infecting humans. Here, we established an in vitro transcription system and identified the composition of Pol II on the viroid RNA template. Interestingly, this remodeled Pol II has a reduced number of components in contrast to the Pol II complex on DNA templates as previously reported. We also analyzed the function of a transcription factor (TFIIIA-7ZF) that aids the Pol II activity on RNA templates. This study provides new insights into Pol II function and viroid infection as well as a robust experimental system for future investigations.

## Introduction

Viroids are single-stranded circular noncoding RNAs that infect crop plants [1,2]. After five decades of studies, the host machinery for viroid infection has not been fully elucidated [1–3]. There are two viroid families, *Avsunviroidae* and *Popspiviroidae* [4,5]. Members of *Pospivoirdae* replicate in the nucleus via the rolling circle mechanism (S1 Fig) and rely on the enzymatic activity of DNA-dependent RNA polymerase II (Pol II) [1–3]. Specifically, ample evidence supports that Pol II activity is critical for the synthesis of oligomer minus-strand or (-) intermediates using viroid circular genomic RNAs as templates [1,6]. However, it remains controversial whether Pol II also uses (-) oligomers as templates for transcription [1,6].

By and large, DNA-dependent RNA polymerases (RNA polymerases or DdRPs) catalyze transcription using DNA templates, which is a fundamental process of life. RNA polymerases are facilitated by a group of general transcription factors to achieve highly regulated transcription, from initiation to elongation and then to termination. Taking Pol II as an example, this 12-subunit complex functions in concert with general transcription factors during transcription initiation around DNA promoter regions [7–10]. In previous reports, a reconstituted system for the DNA promoter-driven transcription requires Pol II and five general transcription factors (TFIIB, TFIID, TFIIE, TFIIF, and TFIIH) [11,12]. Interestingly, the 10-subunit Pol II core (without Rpb4/Rpb7 heterodimer) is sufficient for transcription elongation [13,14]. Transcription elongation is also regulated by multiple factors, including TFIIS, TFIIF, PAF1 (RNA-PII-associated factor 1) complex (PAF1-C), FACT complex (histone chaperone), SPT6, etc [15–17].

Since 1974, RNA polymerases have been found to possess intrinsic RNA-dependent RNA polymerase (RdRp) activity to catalyze RNA polymerization using RNA templates [18]. This intrinsic RdRp activity of RNA polymerases was found in bacteria and mammalian cells, as well as exploited by subviral pathogens (i.e., viroids and human hepatitis delta virus (HDV)) for propagation [19–21]. Pol II transcription using viroid or HDV RNA templates can yield RNA products over 1,000 nt in cells, comparable to some products from DNA templates. To ensure such efficient transcription, specific factors are needed. HDV encodes an S-HDAg protein to promote Pol II activity on its RNA template [22]. Using potato spindle tuber viroid (PSTVd) as a model, we showed that an RNA-specific transcription factor (TFIIIA-7ZF with seven zinc finger domains) interacts with Pol II and enhances Pol II activity on circular genomic RNA template [23,24]. However, it remains unclear how S-HDAg or TFIIIA-7ZF functions with Pol II for RNA-templated transcription.

Biochemically reconstituted systems have been successfully used to characterize the required factors and functional mechanisms underlying DNA-dependent transcription [25]. However, reconstituted transcription systems using RNA templates often exhibited poor

activity. For example, all currently available in vitro transcription (IVT) systems using HDV templates have the premature termination issue generating products less than 100 nt [26,27], which may not reflect the transcription process in cells. We recently established an IVT system for PSTVd that can generate longer-than-unit-length products (more than 360 nt) [23,24], mimicking the replication process in cells [28].

Using our IVT system, we first confirmed that Pol II and TFIIIA-7ZF function together in transcribing (-) PSTVd oligomers to (+) oligomers. Interestingly, we found that the Pol II complex remaining on (-) PSTVd RNA template has a distinct composition as compared with the 12-subunit Pol II or the 10-subunit Pol II core, via nano-liquid chromatography-tandem mass spectrometry analysis (nLC-MS/MS). Rpb4, Rpb5, Rpb6, Rpb7, and Rpb9 were absent in the remodeled Pol II. Notably, Rpb9 is responsible for the fidelity of Pol II transcription. Thus, the absence of Rpb9 may explain the much higher mutation rate of viroid RNA-templated transcription catalyzed by Pol II. Several elongation factors for DNA templates, such as the PAF1 complex and SPT6, were also absent in the transcription complex on RNA templates. More importantly, essential general transcription factors (including TFIIA, TFIIB, TFIID, TFIIE, TFIIF, TFIIH, and TFIIS) were all absent in the transcription complex on PSTVd RNA template, clearly demonstrating the distinct regulations between DNA-dependent and RNA-templated transcription. This distinct Pol II retains the catalytic activity to generate (+) PSTVd oligomers with the aid of TFIIIA-7ZF. We further showed that nearly all seven zinc finger domains of TFIIIA-7ZF are critical for function. In particular, the first three zinc fingers are pivotal for binding with RNA templates. Altogether, this IVT system opens the door to further dissecting the mechanism underlying viroid RNA-templated transcription. Our findings also provide new insights into the organization of Pol II complex on RNA templates, which has profound implications for understanding RNA-templated transcription as well as viroid transcription and its high mutation rates.

## Results

PSTVd replication from circular genomic RNA to (-) oligomers and then to (+) oligomers is a continuous process. When identifying the catalyzing enzyme(s) during PSTVd replication, failure to tease apart each step resulted in controversial data, as evidenced by previous reports [29,30]. To understand whether Pol II can catalyze transcription using (-) viroid oligomers, we established a reconstituted in vitro transcription system using partially purified Pol II from wheat germ [24,31] and the (-) PSTVd dimer as RNA template. The experimental flow is summarized in S1 Fig. In brief, in vitro transcribed (-) PSTVd dimer was mixed with partially purified Pol II with or without the supplement of TFIIIA-7ZF proteins expressed in and purified from bacteria. Noteworthy is that our (-) PSTVd dimer RNA contains non-viroid sequences on both ends, as detailed in S2 Fig. We will discuss the presence of non-viroid sequences below.

Based on our previous report [24] and the rolling circle replication model (S3 Fig), the linear (+) PSTVd (i.e., the product from the IVT assay in this report) cannot serve as a template for further transcription. Therefore, this IVT assay only focuses on the transcription step from (-) PSTVd dimer to (+) PSTVd dimer. As shown in Figs 1A and S4, Pol II consistently exhibited a weak activity in transcribing (-) PSTVd template to (+) PSTVd oligomer with a length similar to the template. We then tested the role of the RNA-specific transcription factor (TFIIIA-7ZF) in this transcription reaction by supplying various amounts of TFIIIA-7ZF protein. The reactions were subject to RNA gel blotting analyses with sequence-specific riboprobes. Comparing the results of RNA gel blots for (-) PSTVd dimer template and the (+) PSTVd product from the same samples demonstrated the specificity of the riboprobes as well

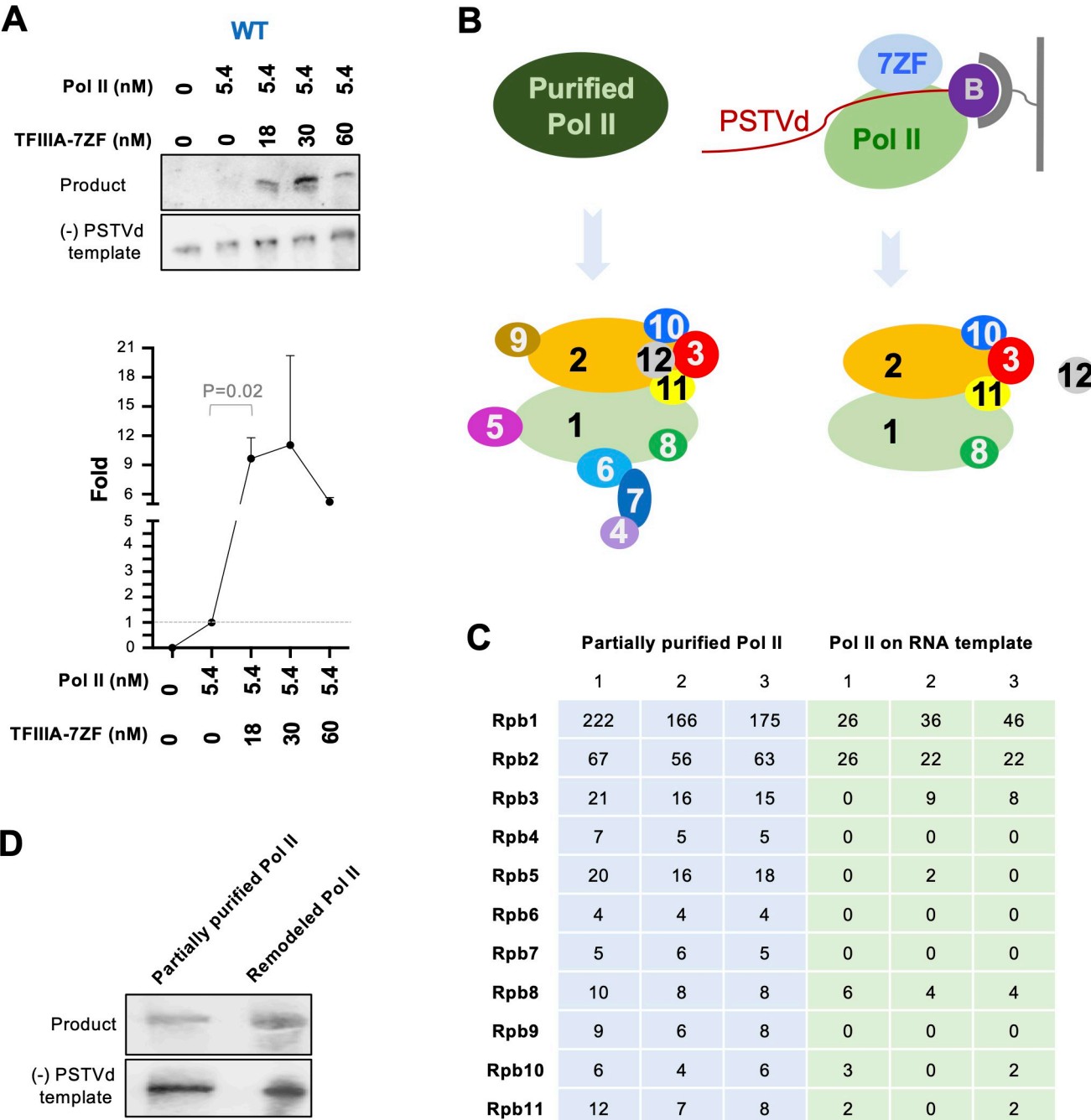

**Fig 1. Uncovering the role of TFIIIA-7ZF and Pol II in transcribing (-) PSTVd oligomers.** (A) Reconstituted in vitro transcription (IVT) system using partially purified Pol II (from wheat germ, 5.4 nM), (-) PSTVd dimer RNA (7.8 nM), and various amounts of TFIIIA-7ZF. Sequence-specific riboprobes were used to detect (-) PSTVd templates and (+) PSTVd products (at the position close to dimer PSTVd). Quantification of product intensities was performed using ImageJ. The first lane signal was set as 0 and the second lane signal was set as 1. Data from three replicates were used for graphing the fold increases induced by various amounts of TFIIIA-7ZF. The uncropped gel images are shown in S4 Fig. (B) Schematic presentation for RNA-based affinity purification followed by nLC-MS/MS identifying protein factors in partially purified Pol II and remodeled Pol II. The presence of Rpb12 in the partially purified Pol II is based on literature. It remains to be determined whether Rpb12 is in the remodeled Pol II. (C) Peptide counts for each Pol II subunit in all nLC-MS/MS replicates. The summarized nLC-MS/MS data are listed in S1 Table. The original data can be found in S1–S6 Datasets. (D) IVT assay demonstrating the activity of remodeled Pol II. The first lane contains free RNA as template, while the second lane contains mixed free RNA and desthiobiothnylated RNA as template. The reaction condition is described in Methods in detail.

as confirmed the identity of the IVT products (Figs 1A and S4). The product signals from samples with Pol II but without TFIIIA-7ZF (Pol II only) were quantified and set as 1. Then the product signals from samples with various amounts of TFIIIA-7ZF protein were normalized to the signal from "Pol II only" sample in the same blot. As shown in Figs 1A and S4, TFIIIA-7ZF can significantly increase the RdRp activity (more than 10-fold) of Pol II on the (-) PSTVd dimer template, which is statistically significant (P value 0.02). Therefore, our results indicate that Pol II can accept (-) PSTVd oligomers as a template for transcription, expanding the known natural RNA templates for DdRPs.

We then analyzed the composition of the Pol II complex on the viroid RNA template using RNA-based affinity purification. Briefly, a desthiobiotinylated cytidine (Bis)phosphate was ligated to the 3' end of (-) PSTVd dimer, which was then mounted to magnetic streptavidin beads. After sequential supplying of TFIIIA-7ZF and partially purified Pol II, the magnetic beads were washed before elution. Through silver staining, common patterns and distinct bands can be observed between elution fraction and partially purified Pol II (S5 Fig), implying that certain factors may be enriched by or removed from RNA templates. We then performed nLC-MS/MS to reveal the protein factors in partially purified Pol II as well as the Pol II complex remaining on the RNA template (proteins identified in each replicate are listed in S1–S6 Datasets). We identified 11 out of 12 Pol II subunits in partially purified Pol II with high confidence (false discovery rate below 0.01, identified by a minimum of 2 peptides, and 2 peptide-spectrum matches) in all three replicates (Fig 1B and 1C and S1 Table). The smallest subunit Rpb12 (~6 kDa) was absent, which is possibly caused by sample loss during the size cut-off enrichment of samples for nLC-MS/MS. This issue has been seen in another study [17]. Interestingly, only six subunits were confidently identified in the Pol II complex remaining on the RNA template with high confidence: Rpb1, Rpb2, and Rpb8 were found in all three replicates while Rpb3, Rpb10, and Rpb11 were found in two out of three replicates (Fig 1C and S1 Table). Rpb5 can only be detected in one replicate of Pol II remaining on the RNA template (Fig 1C). Therefore, we consider it less likely to participate in the Pol II complex on the RNA template. Rpb1 and Rpb2 form the catalytic core of Pol II [32]. Rpb3, Rpb10, and Rpb11 are components of a subassembly group critical for Pol II assembly [32,33]. Rpb8 is an auxiliary factor [32]. Given that the Pol II complex remaining on the RNA template has a distinct composition, we termed it remodeled Pol II hereafter. To test whether the remodeled Pol II has any catalytic activity, we repeated the RNA-based affinity purification followed by IVT. As shown in Fig 1D, this remodeled Pol II indeed possessed transcription activity in generating (+) PSTVd oligomers comparable to partially purified Pol II.

Besides the Pol II complex in the partially purified samples, we also found the presence of several general transcription factors and transcription elongation factors for DNA-dependent transcription. However, they were all absent in the remodeled Pol II in the nLC-MS/MS analysis. For example, we found TFIIF in all three repeats of partially purified Pol II but could not confidently detect it in any of the remodeled Pol II samples (Fig 2 and S1 Table). In addition, TFIIE was found in two out of three repeats of partially purified Pol II but could not be confidently detected in any of the remodeled Pol II samples (S1 Table). Therefore, TFIIE and TFIIF are likely not required for viroid RNA-templated transcription. Noteworthy is that all the rest of the canonical general transcription factors (including TFIIA, TFIIB, TFIID, TFIIH, and TFIIS) were absent in our partially purified or remodeled Pol II, indicating that they are dispensable for (-) PSTVd RNA-templated transcription. SPT6 is a histone chaperone interacting with the Rpb4/Rpb7 heterodimer during transcription elongation on DNA templates [16]. All Rpb4/Rpb7 and SPT6 were absent in the remodeled Pol II samples (Fig 2 and S1 Table). The FACT complex, including SPT16 and SSRP1-B, is also a histone chaperone assisting Pol II elongation on DNA templates. The FACT complex does not interact with Pol II directly

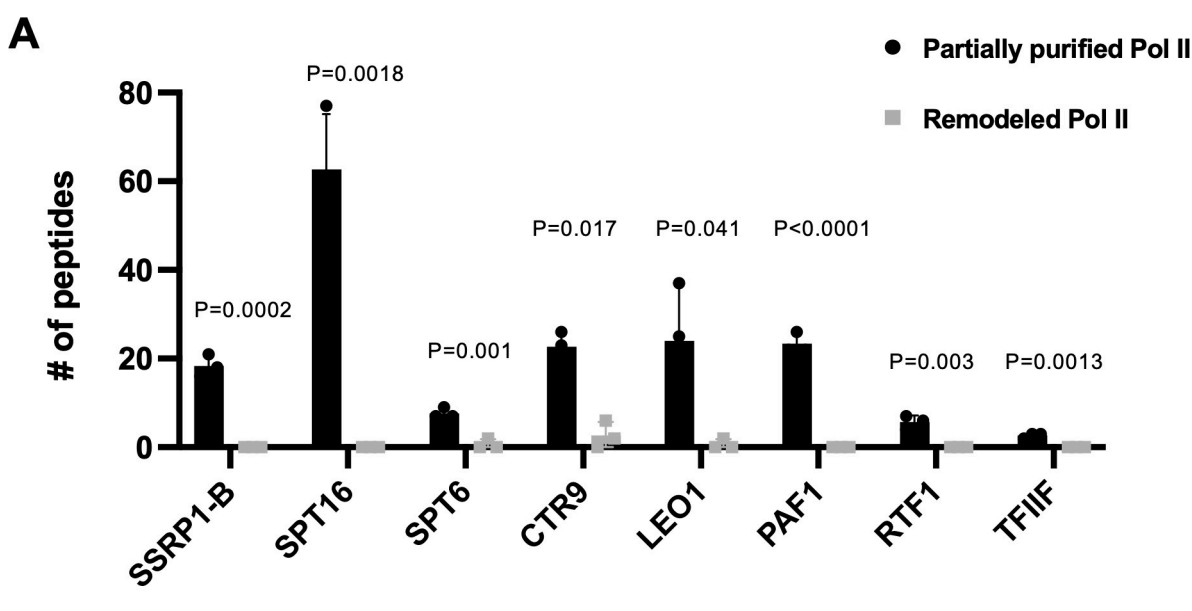

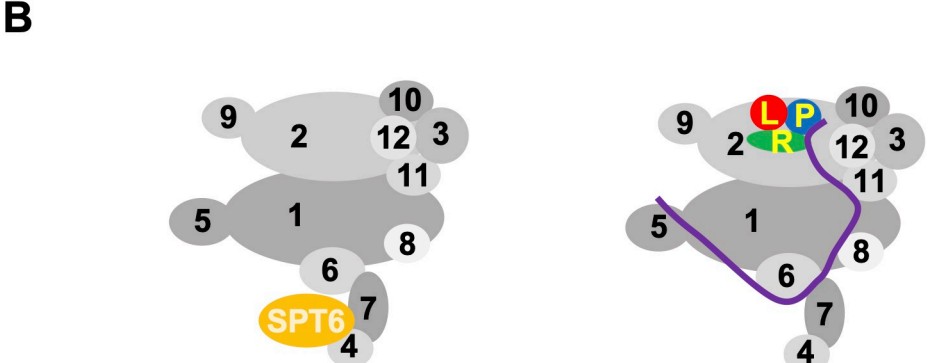

**Fig 2. Analyses on transcription elongation factors.** (A) Analyses of peptide counts of the FACT complex (SSRP1-B, SPT16), SPT6, PAF1-C, and TFIIF in partially purified Pol II and remodeled Pol II. Y-axis lists numbers of identified peptides corresponding to each gene. P values were calculated via a two-tailed T-test, a built-in function in Prism. The summarized nLC-MS/MS data are listed in S1 Table. The original data can be found in S1–S6 Datasets. (B) Schematic presentation of Pol II interactions with SPT6 and PAF1-C during DNA-dependent transcription, based on a reference [16]. P, PAF1. L, LEO1. R, RTF1. Purple line, CTR9.

during transcription on chromatin in cells [34]. Neither of its components was present in the remodeled Pol II samples (Fig 2 and S1 Table). The PAF1-C, including PAF1, CTR9, LEO1, and RTF1, regulates transcription-coupled histone modifications. PAF1-C components extensively interact with Pol II during transcription elongation on DNA template. PAF1-LEO1 is anchored to the external domains of Rpb2. RTF1 is in proximity to PAF1. The trestle helix in CTR9 binds to Rpb5 and the surrounding region, while the tetratricopeptide repeats interact with Pol II around Rpb11 and Rpb8 [16,35]. Similarly, the PAF1-C was mostly absent in the remodeled samples on RNA template (except for the significantly reduced CTR9 found in two out of three replicates) (Fig 2 and S1 Table).

Since TFIIIA-7ZF is critical for Pol II to perform transcription using RNA templates, we attempted to identify the functional domain(s) of TFIIIA-7ZF. TFIIIA-7ZF has seven C2H2 type zinc finger domains. We mutated each zinc finger domain by changing the first histidine

in the C2H2 domain to asparagine, which has been commonly used to disrupt the local structure of a C2H2 motif [36,37]. We then used those variants for the IVT assay. As shown in Figs 3 and S4, all mutants exhibited greatly reduced activity in aiding Pol II transcription on viroid RNA templates. Mutants *zf1*, *zf2*, *zf3*, and *zf6* completely lost the activity, while mutants *zf4*, *zf5*, and *zf7* can increase Pol II activity about two-fold (Fig 3), which is a much weaker activity as compared with more than 10-fold increase stimulated by wildtype (WT) TFIIIA-7ZF (Fig 1A). Despite the low amounts of products generated, products all had a similar length to the template (S4 Fig). We then performed the RNA-based affinity purification assay using TFIIIA-7ZF mutants. Since Pol II itself has viroid RNA binding affinity [38], it is not surprising to observe no difference in the amount of the remodeled Pol II on RNA templates with or without the presence of WT TFIIIA-7ZF (Fig 4). Notably, *zf1*, *zf2*, and *zf3* exhibited significantly reduced affinity to (-) PSTVd dimer RNA, which led to significantly reduced Pol II remaining on RNA templates. The amount of the remodeled Pol II was also reduced, to a lesser extent, in the presence of *zf4*, *zf5*, or *zf7*, which explains the reduced transcription activity in the corresponding IVT reactions. Interestingly, *zf5* had a similar binding ability to RNA template resembling WT and the *zf6* variant, but the amount of Pol II on RNA template was significantly decreased in the presence of *zf5* as compared with the presence of WT or *zf6* (Fig 4). This observation suggests that the fifth zinc finger domain is possibly involved in Pol II binding.

## Discussions

Using a robust IVT platform, we found that a remodeled Pol II and TFIIIA-7ZF can efficiently utilize (-) PSTVd dimer for RNA-templated transcription. TFIIIA-7ZF significantly enhances Pol II transcription activity on RNA templates. This remodeled Pol II represents a new minimal organization of functional Pol II complex. In this remodeled Pol II, we observed the catalytic core (Rpb1 and Rpb2), a subassembly group (Rpb3, Rpb10, Rpb11), and an auxiliary factor Rpb8. Rpb12 was not detected in our samples due to technical shortcomings. Since Rpb12 is a conserved subunit in the subassembly group containing Rpb3, Rpb10, and Rpb11 [32,33], we speculate that Rpb12 is also present in the remodeled Pol II complex. However, it awaits clarification in the future. Rpb4/Rpb7 heterodimer, Rpb6, Rpb9, and likely Rpb5 were absent in the remodeled Pol II. The Rpb4/Rpb7 heterodimer is neither essential for elongation nor included in the Pol II core [13, 14]. Rpb6 is involved in contact with TFIIS and TFIIH [39, 40], neither of which was present in the transcription complex on the (-) viroid RNA template. This is in line with our recent observation that TFIIS is dispensable for PSTVd replication [24]. Rpb9 is critical for Pol II fidelity by delaying NTP sequestration [41, 42]. Pol II fidelity is also partially regulated by TFIIS [43,44]. Interesting, both Rpb9 and TFIIS were absent in the remodeled Pol II samples, which could explain the observation that nuclear-replicating viroids have a much higher mutation rate than cellular Pol II transcripts [45]. Rpb5 has been proposed to make contacts with DNA promoters and coordinate the opening/closing of the Pol II DNA cleft [46–50], which is unlikely involved transcription on single-stranded RNA templates.

It has been proposed that an RNA polymerase may have evolved to transcribe RNA templates first and then transitioned to use DNA templates in modern life forms [51]. Interestingly, Rpb4/Rpb7 heterodimer is present in all archaeal and eukaryotic cells but not in bacteria [10,52], further suggesting that organization changes occurred in RNA polymerases during the course of evolution. The remodeled Pol II identified here only retains a minimal set of factors, while most of the missing subunits are absent in bacterial RNA polymerases (i.e., Rpb4, Rpb5, Rpb7, and Rpb9) [53–55]. Our discovery of a remodeled Pol II actively transcribing RNA templates may provide a handle to further explore the functional evolution of transcription machinery.

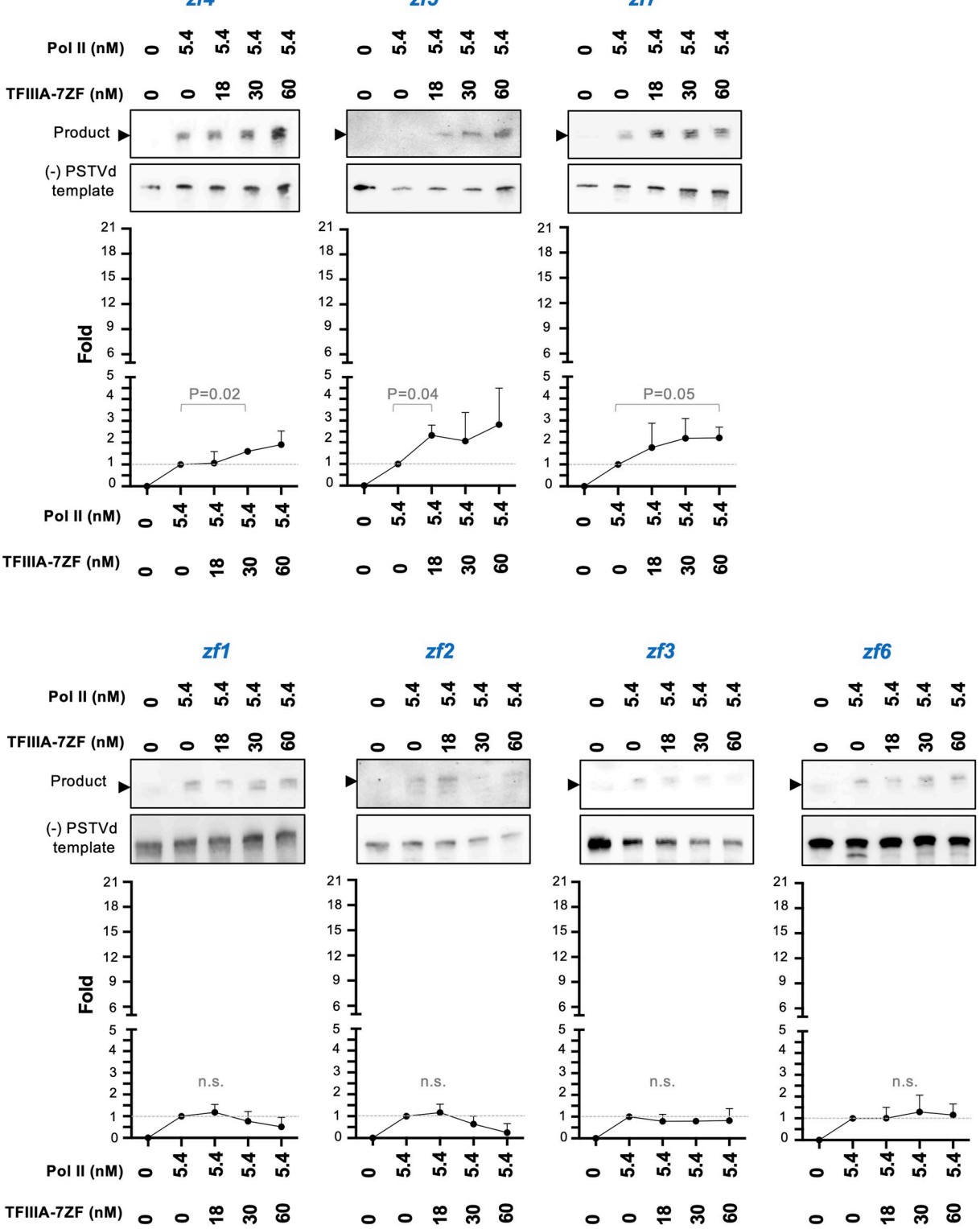

**Fig 3. Analyses on the role of TFIIIA-7ZF zinc finger domains in aiding Pol II activity on RNA templates.** Reconstituted in vitro transcription (IVT) system using partially purified Pol II (5.4 nM), (-) PSTVd dimer RNA (7.8 nM), and various amounts of TFIIIA-7ZF mutants. Sequence-specific riboprobes were used to detect (-) PSTVd templates and (+) PSTVd products. Arrowheads indicate the position of (-) PSTVd dimer template, based on the extremely low cross-reaction signals that were only visible after over-exposure. Fold changes were analyzed as described in Fig 1. P values for significant fold changes as compared with Pol II only samples were listed. n.s., no significant change was identified. Uncropped gel images were shown in S4 Fig.

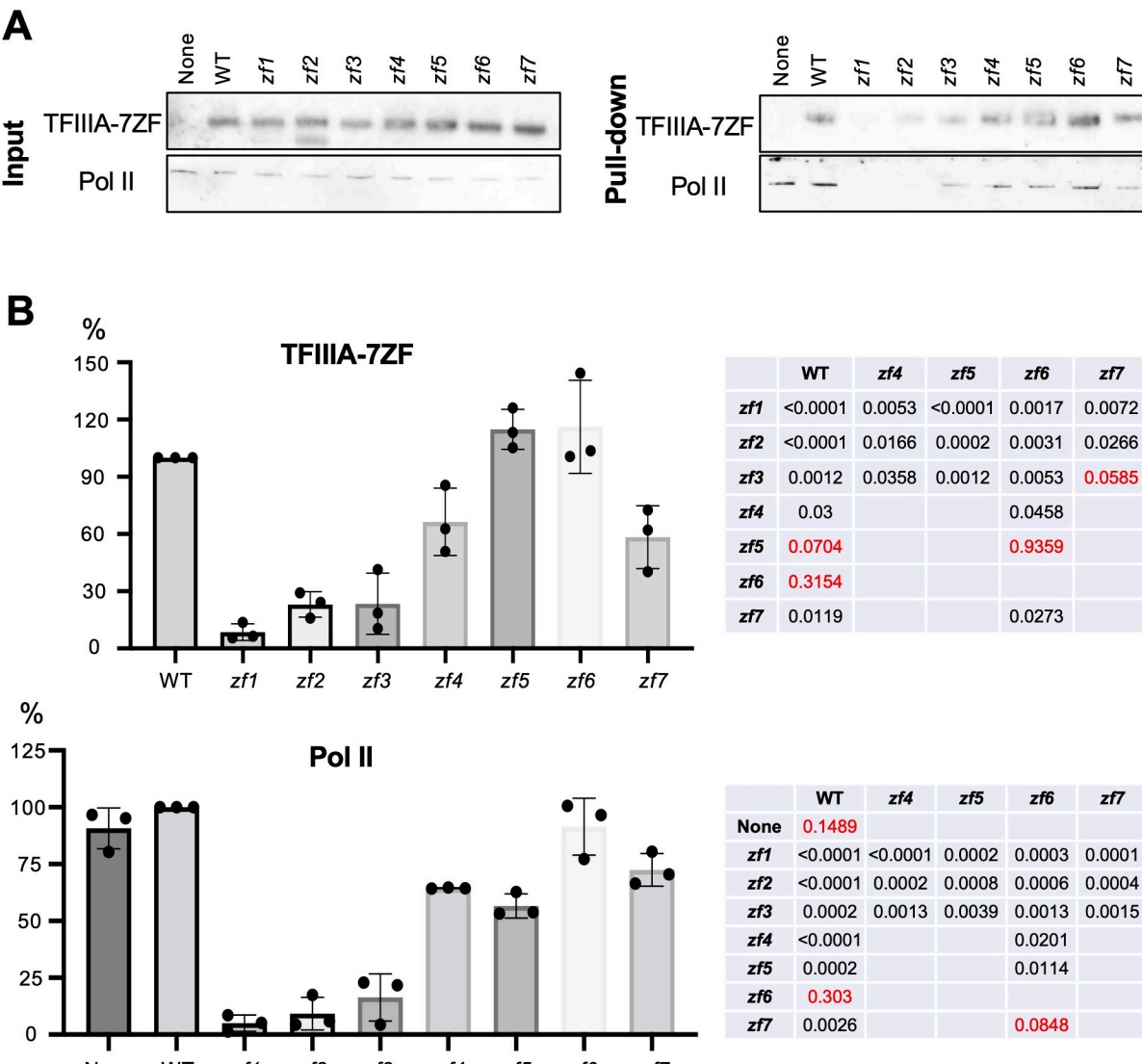

**Fig 4. Analyses of the binding ability of TFIIIA-7ZF mutants.** WT and mutants TFIIIA-7ZF proteins were used for RNA-based affinity purification in the presence of partially purified Pol II. (A) Immunoblots for input and RNA-bound TFIIIA-7ZF (anti-TFIIIA) and the Rpb1 subunit of Pol II (8WG16). Values from the samples with WT TFIIIA-7ZF were set as 100%. Data from samples with mutant TFIIIA-7ZF were normalized to the values of samples with WT TFIIIA-7ZF. None, no TFIIIA-7ZF protein was supplied. (B) Quantification of immunoblotting results in (A). Protein signals in Pull-down blots were normalized to the corresponding signals in the Input blots. The normalized signals in WT (TFIIIA and Rpb1) were set as 100%. Three replicates were performed for quantification and statistical analyses. A two-tailed T-test was used to calculate P values (listed in tables), by using the built-in function in Prism.

The high-resolution crystallographic structure of Pol II core-Rpb4/Rpb7-TFIIS showed that Pol II utilizes the same active site for interacting DNA and RNA templates, revealed by using chimeric RNA templates [51]. Later, one study using a chimeric RNA template containing an HDV fragment sequence suggested that multiple general transcription factors (TFIIA, TFIIB, TFIID, TFIIE, TFIIF, TFIIH, and TFIIS) may potentially be involved in the initiation of RNA-templated transcription [56]. However, neither experimental system could yield long RNA products, suggesting that those transcription complexes might not be optimal for subviral RNA templates. In the transcription complex on PSTVd RNA template, we could not detect the presence of TFIIE or TFIIF, despite that they were both identified confidently in at least

two out of three replicates in partially purified Pol II (S1 Table). Notably, other Pol II-associated general transcription factors (TFIIA, TFIIB, TFIID, TFIIH, and TFIIS) were also absent in partially purified Pol II and remodeled Pol II samples. Although we cannot rule out the possibility that a minute amount of general transcription factors remained in our samples below the detection capacity of nLC-MS/MS, it is unlikely for them to play a role in a stoichiometric ratio to Pol II resembling the transcription complex on DNA templates. Therefore, our observation argues that those general transcription factors for DNA-dependent transcription are dispensable in the transcription complex on RNA templates, at least for (-) PSTVd RNA templates. This is in drastic contrast to the requirement for the formation of a preinitiation complex (PIC) on DNA templates [57]. Thus, our results outlined a distinct organization of transcription complex on viroid RNA templates.

All seven zinc finger domains of TFIIIA-7ZF are critical for the function in aiding RNA-templated transcription. The first three zinc finger domains are pivotal for RNA template binding. Interestingly, Pol II exhibited weaker affinity to RNA templates in the presence of either *zf1*, *zf2*, or *zf3* mutant. It is intuitive to speculate that free *zf1*, *zf2*, and *zf3* proteins sequestered Pol II and prevented Pol II from binding to RNA templates, leading to greatly reduced transcription activity. While *zf5* mutant has a similar ability in binding RNA templates as WT and *zf6*, Pol II was significantly decreased on RNA template in the presence of *zf5* in comparison with the presence of WT or *zf6*. Thus, the fifth zinc finger domain is possibly involved in Pol II binding. It is unclear why the *zf6* mutant also greatly diminished Pol II activity in generating the oligomer product as the amount of *zf6* and Pol II remaining on the RNA template resembles that in the reaction with WT TFIIIA-7ZF. Since this assay only tested TFIIIA-7ZF and Pol II binding to the RNA template before reaction initiates, reasonable speculation is that *zf6* might be critical for transcription initiation or even elongation.

Of note, TFIIIA-7ZF aids Pol II activity on circular (+) PSTVd RNA template as well [23, 24]. The optimized stoichiometric ratios among Pol II, TFIIIA-7ZF, and RNA templates are slightly different between the IVT assays for circular (+) PSTVd [24] and (-) PSTVd dimer templates (reported here). This is likely attributed to the higher amount of circular (+) RNA template used for IVT, which is in agreement with the common observation that circular (+) PSTVd accumulates to a much higher level in vivo as compared with the (-) PSTVd oligomers. It may be informative to compare the organizations of the Pol II complex on distinct RNA templates. However, as we reported recently [24], the Pol II/TFIIIA-7ZF complex does not have any detectable activity to transcribe linear (+) PSTVd RNA template. Therefore, we cannot use the labeled linear (+) PSTVd RNA to purify Pol II for nLC-MS/MS analyses.

Our reconstituted IVT system is robust for exploring the factors and functional mechanism underlying viroid RNA-templated transcription, particularly for studying transcription initiation and elongation. The IVT system reported here paves the way to dissect the machinery and mechanism involved in viroid RNA-templated transcription. One immediate question is whether there is an RNA promoter residing in the (-) PSTVd dimer. Our (-) PSTVd dimer contains a very short non-viroid sequence at its 3' end (6 nt), which falls in the region for the initiation of Pol II transcription. This 6 nt non-viroid sequence appears not to affect Pol II initiation, which is in line with the observation that (-) hop stunt viroid dimer surrounded by non-viroid sequences on both ends remains infectious in plants [58]. Therefore, it is reasonable to state that there is a certain element(s) within (-) viroid oligomers guiding Pol II initiation. In the meanwhile, it is interesting that Pol II seems to initiate transcription only from the 3' end of the RNA template, despite that there is an identical viroid sequence in the middle of the oligomer RNA template. As evidence, we could not observe any smaller products in RNA gel blots (S4 Fig). We expect our IVT system can help future investigations on these topics.

Notably, the mechanism underlying transcription termination on viroid RNA template remains unknown [3]. Future investigations using our reconstituted IVT system may help analyze the transcription termination mechanism (regulated or run-off) and provide a handle for experimental tests in plants. In addition, the detailed regulation and mechanism for the remodeled Pol II/TFIIIA-7ZF complex on viroid RNA templates remains to be elucidated, which may be addressed by structural analyses in the future.

Although IVT systems are powerful to dissect the machinery and mechanism underlying the biological processes that are difficult to analyze in vivo, they may introduce some variations. For example, the requirement of general transcription factors for the formation of PIC in reconstituted assays is largely in the agreement with data obtained from in vivo studies but with minor variations [57]. On the other hand, the FACT complex and PAF1 complex regulate transcription elongation in cells but are dispensable for in vitro transcription [13,14], which may be attributed to the lack of histone-binding for in vitro DNA templates. Therefore, despite that we did not observe the FACT components and most of the PAF1 complex (except for the significantly reduced CTR9) on RNA template (Fig 2 and S1 Table), we cannot rule out the possibility that they might function in the cellular environment. While the finding of a minimal organization of Pol II complex with a reduced set of factors catalyzing (-) PSTVd RNA-templated transcription is novel, developing an in vivo experimental system is desirable to corroborate this finding. Such investigations need to overcome the challenge of distinguishing the remodeled Pol II on viroid RNA from the dominant Pol II complex/subunits in association with DNA in cells.

## Materials and methods

### Molecular constructs

We have previously reported WT TFIIIA-7ZF cloned from *Nicotiana benthamiana* in bacteria expression vector pTXB1 (New England Biolabs, Ipswich, MA) [23]. The TFIIIA-7ZF mutants were generated via site-directed mutagenesis using the WT TFIIIA-7ZF in pTXB1 as the template (See S2 Table for primer information). We have also reported the pInt95-94(-) and pInt95-94(+) constructs for generating PSTVd probes to detect sense and antisense PSTVd RNAs, respectively [23]. PSTVd dimer construct was inherited from late Professor Biao Ding (Ohio State University) and was illustrated in S2 Fig. All constructs have been verified by Sanger sequencing.

### Protein purification

The protocol for recombinant protein purification was based on our reported protocol [23]. Various recombinant TFIIIA-7ZF proteins with an intein-chitin binding domain (CBD) tag were overexpressed using the *Escherichia coli* BL21(DE3) Rosetta strain (EMD Millipore, Burlington, MA). For each construct, about 500 mL of bacterial culture was collected and re-suspended. After sonication with Bioruptor (Diagenode, Denville, NJ), samples were subjected to centrifugation at 15,000X $g$ for 1 h at 4˚C. The cell lysate was collected and incubated for 1 h with 2 mL of 50% slurry of chitin resin (New England Biolabs) before loading onto an empty EconoPac gravity-flow column (Bio-Rad Laboratories, Hercules, CA). After washing, the resin was incubated for 18 h at 4˚C in a cleavage buffer [20 mM Tris-HCl (pH 8.5), 500 mM NaCl, 50 mM dithiothreitol, and 5 μM $ZnSO_4$]. Fractions containing tagless proteins were dialyzed against 20 mM HEPES, pH 7.5, 200 mM NaCl, 50 μM $ZnSO_4$, and 5 mM DTT. Protein concentrations were estimated by Coomassie Brilliant Blue staining of an SDS-PAGE gel using bovine serum albumin of known concentrations as reference standards.

Purification of Pol II from wheat germ was carried out following our published protocol [24]. All operations were performed at 4˚C, and all centrifugations were carried out for 15 min. One hundred grams of raw wheat germ (Bob's Red Mill, Milwaukie, OR) was ground in a Waring Blender with 400 mL of buffer A [50 mM Tris-HCl pH 7.9, 0.1 mM EDTA, 1 mM DTT, and 75 mM $(NH_4)_2SO_4$]. The resulting homogenate was diluted with 100 mL of buffer A and followed by centrifugation at 15,000X $g$. The supernatant was filtered through one layer of Miracloth (MilliporeSigma, Burlington, MA). The resulting crude extract containing Pol II was precipitated by an addition of 0.075 volume of 10% (v/v) Polymin P with rapid stirring. The resulting mixture was subject to centrifugation at 10,000X $g$. The pellet was washed with 200 mL of buffer A. The insoluble fraction, which contains Pol II, was resuspended with buffer B [50 mM Tris-HCl pH 7.9, 0.1 mM EDTA, 1 mM DTT, and 0.2 M $(NH_4)_2SO_4$]. The resulting suspension was centrifuged at 10,000X $g$ to remove insoluble pellets. $(NH_4)_2SO_4$ precipitation was carried out by slowly adding 20 g of solid $(NH_4)_2SO_4$ per 100 mL of the above supernatant with stirring. The mixture was centrifuged, and the pellet was dissolved in buffer C (0.05 M Tris-HCl pH 7.9, 0.1 mM EDTA, 1 mM DTT, 25% ethylene glycol) plus 0.1% Brij 35 (Thermo Fisher Scientific, Waltham, MA) to make final $(NH_4)_2SO_4$ concentration 0.15 M. The $(NH_4)_2SO_4$ concentration was determined by conductivity. The resulting solution was applied to DEAE Sepharose FF (GE Healthcare Life Sciences, Pittsburgh, PA) equilibrated with buffer C plus 0.15 M $(NH_4)_2SO_4$. The column was then washed with five-bed volumes with buffer C containing 0.15 M $(NH_4)_2SO_4$. Finally, bound Pol II was eluted with buffer C containing 0.25 M $(NH_4)_2SO_4$. Fractions containing Pol II were pooled. The $(NH_4)_2SO_4$ concentration was adjusted to 75 mM by conductivity. The resulting solution was applied to SP Sepharose FF (GE Healthcare Life Sciences) equilibrated with buffer C containing 75 mM $(NH_4)_2SO_4$. After washing the column with the same buffer, Pol II was eluted using the buffer C containing 0.15 M $(NH_4)_2SO_4$. Eluted fractions containing Pol II were pooled. Ethylene glycol (VWR Chemicals BDH, Radnor, PA) was added to a final concentration of 50% (v/v) before storing at −20˚C.

## In vitro transcription assay

Pol II-catalyzed in vitro transcription was carried out based on our recently developed protocol [23, 24]. BSA (New England Biolabs) and TFIIIA-7ZF were treated with 1 unit of Turbo DNase (Thermo Fisher Scientific) for 10 min at 37˚C. Then, 0.39 pmol (-) PSTVd dimer, 0.27 pmol partially purified Pol II, pretreated BSA (4 μM final concentration), and various amounts of TFIIIA-7ZF were incubated at 28˚C for 15 min. The reaction system was adjusted to contain 50 mM HEPES-KOH pH 7.9, 1 mM $MnCl_2$, 6 mM $MgCl_2$, 40 mM $(NH_4)_2SO_4$, 10% (v/v) glycerol, 1 unit/μL SuperaseIn RNase inhibitor (Thermo Fisher Scientific), 0.5 mM rATP, 0.5 mM rCTP, 0.5 mM rGTP, 0.5 mM rUTP. Transcription reactions (50 μL) were incubated at 28˚C for 4 h. About 0.8 U/μL proteinase K (New England Biolabs) was applied to terminate the reaction by incubation at 37˚C for 15 min, followed by incubation at 95˚C for 5 min. The Pol II-catalyzed in vitro transcription assay was repeated three times for each TFIIIA-7ZF variant. For the IVT assay in Fig 1D, TFIIIA-7ZF and partially purified Pol II were subsequentially bound to immobilized desthiobiotinylated RNA templates (see the section below for details). After washing twice, additional RNA templates without desthiobiotinylation were supplied together with NTPs. The reaction was then performed the same as aforementioned. This assay was repeated twice.

## RNA-based affinity purification

Using Pierce RNA 3' end desthiobiotinylation kit (Thermo Fisher Scientific, Waltham, MA, USA), 50 pmol of PSTVd dimer RNA was labeled following the manufacturer's instructions.

Labeled RNA was purified using the MEGAclear kit (Thermo Fisher Scientific, Waltham, MA, USA) and heated at 65˚C for 10 min followed by incubation at room temperature for 12 min. Labeled RNA was bound to the 50 μL of streptavidin magnetic beads (Thermo Fisher Scientific, Waltham, MA, USA). Magnetic beads were collected and washed twice with an equal volume of 20 mM Tris-HCl, pH 7.5. Beads were subsequently washed with reaction buffer containing, 50 mM HEPES-KOH pH 8, 2 mM $MnCl_2$, 6 mM $MgCl_2$, 40 mM $(NH_4)_2SO_4$, and 10% glycerol. DNase-treated 150 pmol of recombinant TFIIIA-7ZF was incubated with RNA-bound beads in a 50 μL reaction at 28˚C for 15 min. Then, 27 pmol of partially purified Pol II was added to the reaction and incubated at 28˚C for another 15 min. Next, beads were washed twice with wash buffer (20 mM Tris-HCl, pH 7.5, 10 mM NaCl, 0.1% Tween-20) and bound proteins were eluted with 1X SDS-loading buffer by heating 95˚C for 5 min.

## RNA gel blots

The detailed protocol has been reported previously [59,60]. Briefly, after electrophoresis in 5% (w/v) polyacrylamide/8 M urea gel for 1 h at 200 V, RNAs were then transferred to Hybond-XL nylon membranes (Amersham Biosciences, Little Chalfont, United Kingdom) by a Bio-Rad semi-dry transfer cassette and were immobilized by a UV-crosslinker (UVP, Upland, CA). The RNAs were then detected by DIG-labeled UTP probes. PSTVd RNAs were prepared as described before [23]. *Sma*I-linearized pInt95-94(-) and pInt95-94(+) were used as templates for generating probes, using the MAXIscript kit (Thermo Fisher Scientific). The DIG-labeled probes were used for detecting PSTVd RNAs. RNA gel blot signals were obtained using ChemiDoc (Bio-Rad Laboratories) and quantified using ImageJ (https://imagej.nih.gov/ij/). The quantified signals with biological replicates were subject to graphing and statistical analysis using the built-in functions of Prism (GraphPad Software, LLC). The raw quantification data used for graphing were listed in S7 Dataset.

## Immunoblots

Protein samples were separated on an SDS-PAGE gel, followed by transfer to nitrocellulose membrane (GE Healthcare Lifesciences) using the Mini-PROTEAN Tetra Cell (Bio-Rad Laboratories). After 1 h incubation with 1% (w/v) nonfat milk in 1X TBS (50 mM Tris-HCl, pH 7.5, 150 mM NaCl) at room temperature, membranes were incubated with primary antibodies overnight at 4˚C. After three washes with 1X TBST (50 mM Tris-HCl, pH 7.5, 150 mM NaCl, 0.1% Tween 20), HRP-conjugated secondary antibodies were added. Membranes were then washed three times with 1X TBST and incubated with HRP substrates (Li-COR Biosciences, Lincoln, NE). The signals were detected with ChemiDoc and quantified using ImageJ. Graphing and statistical analyses were performed using the built-in functions of Prism. The raw quantification data used for graphing were listed in S7 Dataset.

For immunoblotting, polyclonal antibodies against TFIIIA were diluted as 1:2,000 and the monoclonal 8WG16 antibodies (Thermo Fisher Scientific) were diluted at 1:1,000. HRP-conjugated anti-mouse serum (Bio-Rad) was diluted at 1:5,000. HRP-conjugated anti-rabbit serum (Thermo Fisher Scientific) was diluted at 1:3,000. For silver staining, we followed the instructions of the Silver BULLit kit (Amresco, Solon, OH).

## Nano-liquid chromatography-tandem mass spectrometry analysis (nLC-MS/MS)

Prior to mass spectrometry, samples were subjected to in-solution digestion. Briefly, reduction treatment (100 mM dithiothreitol and 15 min incubation at 65˚C) and alkylation treatment (100 mM iodoacetamide/45 min incubation at room temperature) were followed by 16 h

incubation at 37°C with sequencing grade trypsin (Promega, Madison WI). Tryptic peptides were acidified with formic acid, lyophilized, and stored at -80°C. As described previously [61], two micrograms of digested protein were subjected to nLC-MS/MS analysis using the LTQ-Orbitrap Velos mass spectrometer (Thermo Fisher Scientific) directly linked to the Ultimate 3000 UPLC system (Thermo Fisher Scientific), with the following modification: mass spectra of intact and fragmented peptides were collected in the orbitrap and linear ion trap detector, respectively. All data files were deposited to the PRIDE database [62] (accession PXD033736). The.raw mass spectral files were searched using the SEQUEST algorithm of the Proteome Discoverer (PD) software version 2.1 (Thermo Fisher Scientific). Tolerances were set to 10 ppm and 0.8 Da to match precursor and fragment monoisotopic masses, respectively. The *Triticum aestivum* NCBI Ref protein database (as of February 2022, with 122,221 entries) and its reversed copy served as the target and decoy database, respectively, to allow the calculation of False Discovery Rate (FDR). All proteins presented in the results were filtered by FDR<1% as well as identified by a minimum of 2 peptides and 2 PSMs (peptide-spectrum matches) in each replicate. The PD result data files showing peptide/protein-ID relevant parameters for each individual replicate are given in S1–S6 Datasets.

## Supporting information

**S1 Table. Summary results for transcription related factors.**
(XLSX)

**S2 Table. Primer sequences for cloning.**
(XLSX)

**S1 Fig. Flow chart illustrating the reconstituted in vitro transcription system.**
(TIF)

**S2 Fig. Illustration for the construct and the corresponding (-) PSTVd dimer template.**
(TIF)

**S3 Fig. Rolling circle replication of PSTVd.**
(TIF)

**S4 Fig. Uncropped RNA gel blotting images for Fig 1A and Fig 3.** Arrowheads indicate the position of (-) PSTVd dimer template, based on the extremely low cross-reaction signals that were only visible after over-exposure. The product signals were often retarded as compared with the template, likely due to the high-salt/high-volume loading effect.
(TIF)

**S5 Fig. Silver staining of RNA-based affinity purification fractions.** Pol II subunits are labeled based on the predicted molecular weight.
(TIF)

**S1 Dataset. nLC-MS/MS for partially purified Pol II (replicate 1).**
(XLSX)

**S2 Dataset. nLC-MS/MS for partially purified Pol II (replicate 2).**
(XLSX)

**S3 Dataset. nLC-MS/MS for partially purified Pol II (replicate 3).**
(XLSX)

**S4 Dataset. nLC-MS/MS for remodeledPol II (replicate 1).**
(XLSX)

**S5 Dataset. nLC-MS/MS for remodeledPol II (replicate 2).**
(XLSX)

**S6 Dataset. nLC-MS/MS for remodeledPol II (replicate 3).**
(XLSX)

**S7 Dataset. Numerical values for graphing in Fig 1A, Fig 3, and Fig 4B.**
(XLSX)

## Acknowledgments

We are grateful to Donna Gordon at Mississippi State University for the constructive discussions. We thank Lexie Thomas at Mississippi State University for technical assistance.

## Accession numbers

The raw mass spectrometry proteomics data have been deposited to the ProteomeXchange Consortium via the PRIDE partner repository with the dataset identifier PXD033736. The organized mass spectrometry proteomics data of each replicate are summarized in S1–S6 Datasets.

## Author Contributions

**Conceptualization:** Ying Wang.

**Formal analysis:** Bin Liu, Ying Wang.

**Funding acquisition:** Ying Wang.

**Investigation:** Shachinthaka D. Dissanayaka Mudiyanselage, Junfei Ma, Tibor Pechan, Olga Pechanova.

**Methodology:** Shachinthaka D. Dissanayaka Mudiyanselage, Ying Wang.

**Project administration:** Ying Wang.

**Resources:** Junfei Ma.

**Supervision:** Ying Wang.

**Validation:** Shachinthaka D. Dissanayaka Mudiyanselage, Tibor Pechan.

**Visualization:** Shachinthaka D. Dissanayaka Mudiyanselage, Ying Wang.

**Writing – original draft:** Shachinthaka D. Dissanayaka Mudiyanselage, Ying Wang.

**Writing – review & editing:** Shachinthaka D. Dissanayaka Mudiyanselage, Junfei Ma, Tibor Pechan, Olga Pechanova, Bin Liu, Ying Wang.

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
