## [Decision Letter · Decision Letter 0]

8 Aug 2022

Dear Dr. Ying Wang,

Thank you very much for submitting your manuscript "A remodeled RNA polymerase II complex catalyzing viroid RNA-templated transcription" for consideration at PLOS Pathogens. As with all papers reviewed by the journal, your manuscript was reviewed by members of the editorial board and by several independent reviewers. The reviewers appreciated the attention to an important topic. Based on the reviews, we are likely to accept this manuscript for publication, providing that you modify the manuscript according to the review recommendations.

Two out of three reviewers appreciate this work represents an important advance in viroid replication. I personally agree with this appreciation. My suggestion is that you try to clarify the points raised by the reviewers and that, in the Discussion section, comment on future directions to bring this research line to a more in vivo context, as one of the reviewer is demanding.

Sincerely,

José-Antonio Daròs

Guest Editor

PLOS Pathogens

Peter Nagy

Section Editor

PLOS Pathogens

Kasturi Haldar

Editor-in-Chief

PLOS Pathogens

orcid.org/0000-0001-5065-158X

Michael Malim

Editor-in-Chief

PLOS Pathogens

orcid.org/0000-0002-7699-2064

Two out of three reviewers appreciate this work represents an important advance in viroid replication. I personally agree with this appreciation. My suggestion is that you try to clarify the points raised by the reviewers and that, in the Discussion section, comment on future directions to bring this research line to a more in vivo context, as one of the reviewer is demanding.

Reviewer Comments (if any, and for reference):

Reviewer's Responses to Questions

**Part I - Summary**

Reviewer #1: In this work submitted by Dissanayaka Mudiyanselage et al, is showed (by using an “in vitro” system) that Pol II is able to accept (-)PSTVd RNA template to generate plus-strand vdRNAs. Furthermore, the components of this “in vitro” transcriptionally active complex were analyzed by mass spectrometry to identify their potential components.

The manuscript is well organized and the experimental design is adequate, however the obtained data arise from an artificial system, and the authors cannot determine that similar complex could also exist during viroid infection.

Reviewer #2: This is a well written manuscript that provides relevant evidence on the involvement of RNA polymerase II (Pol II) the in the synthesis of the plus polarity strand of potato spindle tuber viroid (PSTVd), the type member of the viroid family Pospiviroidae, thus addressing and clarifying a still controversial issue regarding the replication of this nuclear viroid through the asymmetric pathway of the rolling circle mechanism. Moreover, the authors showed that the Pol II complex on the viroid RNA template is remodeled with respect to that acting on DNA templates, with several transcription factors reported in the latter lacking in the former complex where a major role to increase the transcription efficiency of the RNA template is played by the transcription factor TFIIIA-7ZF. This manuscript provides an experimental system to further dissect the transcription also of other viroids and viroid-like RNAs, thus being of general interest for the study of these infectious agents and for further dissect the composition of RNA II transcription complexes using the RNA as a template.

Experiments are carefully performed with appropriate replicates and controls

The manuscript deserves publication.

Reviewer #3: This is a well executed study and well written manuscript.

The authors used their IVT system followed by nano-liquid chromatography, and provided evidence that a modified-simplified Pol II complex, aided by TFIIIA-7ZF, can utilize as template (-) PSTVd to produce (+) oligomers.

The study also identified the critical zinc finger domains of the TFIIIA-7ZF for pol II RNA template binding.

This study contributes novel and important information. It performed in vitro experiments with systems that allowed a closer representation of cellular activities (e.g., size of viroid template and product) and provide insights on the adaptation of the DNA dependent RNA pol II to accept as a template (-) sense RNA. The results of the study are also in agreement with the hallmark of the error prone RNA replication as well as the evolutionary movement from the RNA to the DNA world and from prokaryotic to eukaryotic organisms since it identified that (-) sense viroid RNA is utilizing a simplified pol II, lacking a substantial number of sub-units including the ones required for proof reading activity, as well as the absence of several transcription factors that are typically present in eukaryotic cells when DNA is the template for pol II.

In the cases where the study did not produce clear evidence the authors identified shortfalls of the experimental tools and identified the needs for further work.

**Part II – Major Issues: Key Experiments Required for Acceptance**

Reviewer #1: Additionally, some additional controls may contribute to elucidate the transcriptional efficiency of the complex Pol II/PSTVd. For example the use of canonical PSTVd(+) template as comparative transcription control, could be a good parameter to analyze if the activity of this (PSTVd(-)/Pol II) transcriptional complex is comparable or only a residual effect.

On the other hand, internals controls with the WT (TFIIIA-7ZF complex) should be included in the mutants-assays in order to provide an internal control that allowed perform a true comparative analysis of the transcriptional activity between WT and mutants. For example according the figures showed, this reviewer consider that the transcriptional activity of the WT complex (Fig 1A), is comparable or minor that the observed with the mutants zf4 and zf7.

Finally, the figures are poorly explicative, making hard the interpretation of the results. For example, internals size controls are missing, the authors claim about the longitude of the obtained transcripts, but size references (transcripts obtained by conventional transcription from dimeric clones) are no provided.

Reviewer #2: I think that the point reported below should be addressed by the authors to increase the clarity and the completeness of the manuscript:

My major concern regards the dimeric RNA template used in the reported experiments. It has been described in details in a previous publication. However, in the context of this manuscript is it is very important to provide a clear description of such a template and possibly address more in details (maybe in the discussion) several issues that likely will be further investigated in the future.

The authors should clarify somewhere in the main text whether such a template includes non-viroid sequences at its terminal ends. Which are the initial and terminal positions of the used template should also be clarified. Do the authors expect the same transcription efficiency (or hte same size of products) using dimeric templates starting at different positions? Related to this is the question of whether a promoter does exist on this template and whether the authors retain that their system may help to clarify this point.

Another point regards the size of the generated RNA (more than 700 nt).

A first concern is that in the reported figures it is not possible to appreciate the size of the transcript f. A figure representing the complete gel could be provided as supplementary material to appreciate the marker and the absence of additional (maybe shorter?) transcription products.

Another issue, always related to this point, is the conclusion by the authors (Line 267) that in their experimental system the transcription “is terminated by template run-off”. It does not seem that experimental data are provided to support this conclusion. This point should be better clarified, highlighting the data supporting such a conclusion.

Reviewer #3: I do not see any major issues with the study. The experimental systems are well established and this study is building and expanding on previous studies with proven results.

**Part III – Minor Issues: Editorial and Data Presentation Modifications**

Reviewer #1: (No Response)

Reviewer #2: L 25:I would avoid the use of mysterious here. Viroids are quite well characterized infectious agents as stated below.

L29-30: Maybe clarify here that the involvement of Pol II in the synthesis of minus strand has been already addressed and confirmed.

L120: How the size of the transcribed sequences were confirmed? No marker in the figure is shown (the full gel could be provided as supplementary figure. In addition, the fact that the identity was confirmed by Northern blot hybridization is reported in the legend of the figure, I suggest to clarify this also in the main text.

Reviewer #3: 1. This study is building on previous studies of the same group. For a first time reader however, that has not digested the previous studies (e.g., 23, 24) it could be challenging to follow the experiments sequence.

For example, was Pol II purified from wheat once and a portion was studied for its "WT" properties/complex structure and another portion was used in the IVT system with (-) PSTVd template to study the remodeled Pol II complex structure?

OR

Is the IVT system is more of a "kit/off the self" kind of system?

In addition, while Pol II is generated in planta other elements of the study (i.e., transcription factors) are generated in vitro/bacterial systems. This also could be a source of confusion.

Maybe a small schematic/flowchart, perhaps as a supplemental figure, of the experimental approach and how all the different elements, come together and how they are analyzed, can improve the reader's experience.

2. Please check #51 reference. Is it the correct/best one for the PSTVd dimer construction?

PLOS authors have the option to publish the peer review history of their article (what does this mean?). If published, this will include your full peer review and any attached files.

Reviewer #1: No

Reviewer #2: No

Reviewer #3: No

Figure Files:

Data Requirements:

Reproducibility:

References:

---

## [Editor Report · Decision Letter 1]

1 Sep 2022

Dear Dr. Wang,

We are pleased to inform you that your manuscript 'A remodeled RNA polymerase II complex catalyzing viroid RNA-templated transcription' has been provisionally accepted for publication in PLOS Pathogens.

Best regards,

José-Antonio Daròs

Guest Editor

PLOS Pathogens

Peter Nagy

Section Editor

PLOS Pathogens

Kasturi Haldar

Editor-in-Chief

PLOS Pathogens

orcid.org/0000-0001-5065-158X

Michael Malim

Editor-in-Chief

PLOS Pathogens

orcid.org/0000-0002-7699-2064

The manuscript by Dissanayaka Mudiyanselage and collaborators represents an insightful advance about viroid replication. In their revised version, authors have properly addressed all comments and concerns raised by the three reviewers.

---

## [Editor Report · Acceptance letter]

14 Sep 2022

Dear Dr. Wang,

We are delighted to inform you that your manuscript, "A remodeled RNA polymerase II complex catalyzing viroid RNA-templated transcription," has been formally accepted for publication in PLOS Pathogens.

Best regards,

Kasturi Haldar

Editor-in-Chief

PLOS Pathogens

orcid.org/0000-0001-5065-158X

Michael Malim

Editor-in-Chief

PLOS Pathogens

orcid.org/0000-0002-7699-2064